# Development and Validation of a Dual-Language (English and Malay) Needs Assessment Tool for Breast Cancer (NeAT-BC)

**DOI:** 10.3390/diagnostics13020241

**Published:** 2023-01-09

**Authors:** Yek-Ching Kong, Mahmoud Danaee, Ranjit Kaur, Muthukkumaran Thiagarajan, Hafizah Zaharah, Mustafa Sener, Harenthri Devy Alagir Rajah, Nur Aishah Taib, Karuthan Chinna, Cheng-Har Yip, Nirmala Bhoo-Pathy

**Affiliations:** 1Department of Social and Preventive Medicine, Faculty of Medicine, Universiti Malaya, Kuala Lumpur 50603, Malaysia; 2Breast Cancer Welfare Association Malaysia, Petaling Jaya 46200, Malaysia; 3Department of Radiotherapy and Oncology, Kuala Lumpur Hospital, Kuala Lumpur 50586, Malaysia; 4Department of Radiotherapy and Oncology, National Cancer Institute, Putrajaya 62250, Malaysia; 5Erasmus University Medical Centre, 3015 GD Rotterdam, The Netherlands; 6Department of Surgery, Faculty of Medicine, Universiti Malaya, Kuala Lumpur 50603, Malaysia; 7Faculty of Business and Management, UCSI University, Kuala Lumpur 56100, Malaysia; 8Subang Jaya Medical Centre, Subang Jaya 47500, Malaysia

**Keywords:** needs assessment, breast cancer, questionnaire, development, validation, Asia

## Abstract

Background: Needs assessment tools may be used to guide the optimisation of cancer survivorship services. We sought to develop and validate a dual-language needs assessment tool for women with breast cancer. Methods: The study comprised two phases; (I) co-design, and (II) psychometric testing. In Phase I, items were generated based on focus group discussions with patients and a literature review. These items were then translated into the Malay language. Content and face validation were undertaken with an expert panel. In phase II, 315 Malaysian women living with breast cancer were recruited. Exploratory factor analysis (EFA) and confirmatory factor analysis (CFA) were conducted to determine construct validity and reliability. Criterion validity was assessed using the EORTC QLQ-C30 questionnaire. Results: We co-designed and validated a 48-item dual-language needs assessment tool for breast cancer (NeAT-BC). Five underlying constructs were revealed in the EFA: (1) emotional support, (2) social and intimate relationships, (3) hospital appointments, (4) personal care and health, and (5) information and services. The NeAT-BC demonstrated good reliability across all constructs (Cronbach’s alpha: 0.90 to 0.96). CFA also demonstrated acceptable convergent and divergent validity, composite reliability ≥ 0.87, and Heterotrait–Monotrait index < 0.85 for all constructs. Criterion validity was established given the significant negative correlation between overall needs and quality of life (r = −0.14; *p* = 0.02). The NeAT-BC took approximately 25 min to be completed and could be interviewer-administered or self-administered. Conclusion: The utilization of the NeAT-BC is expected to guide establishment of evidence-based cancer survivorship services in Malaysia, with wider potentials for adoption in other multi-ethnic and/or low-and-middle income settings.

## 1. Introduction

Improvement in survival rates and the resulting growing population of breast cancer survivors have brought about an increased awareness of the importance of addressing their supportive care needs that goes beyond the provision of cancer therapies alone [1]. Ideally, cancer survivorship programs should provide essential services that address wider aspects of wellbeing, including provision of emotional, informational, social, spiritual, and financial supportive services [2]. Equally important is that such services also cater to the differing needs of survivors as they transition along the cancer care continuum [3]. Notably, cancer survivors who reported having their needs sufficiently met tend to report a better perceived health-related quality of life and experience less psychological distress [4,5].

Compared to their counterparts with cancer in other sites, women living with breast cancer were more likely to report having higher unmet needs, particularly in dealing with the physical side effects of treatments, financial issues, and navigating the healthcare system [6]. Apart from individual socio-demographic characteristics, the needs of patients following a cancer diagnosis are also influenced by the national health and social support systems in place, as well as the broader socio-cultural beliefs of the population. For instance, a study comparing unmet needs among breast cancer survivors living in Hong Kong and Germany described significant cultural-specific disparities in the types of unmet supportive care needs that were prioritized by the two different populations of women [7]. Similarly, another study comparing Japanese, Hong Kong Chinese, and Taiwanese Chinese cancer survivors found that even within the Asian population, interactions between socio-cultural factors and healthcare services were more strongly associated with the differences in needs between the three groups of survivors compared to clinical and demographic factors [8].

Malaysia is a multi-ethnic, multi-cultural, middle-income Asian country that has made good progress toward universal health coverage via a mixed public and private healthcare delivery system [9]. Although cancer care is currently available at highly subsidized rates via a nationwide network of public hospitals throughout the country, a report on breast cancer by The Economist Intelligence Unit in 2016 had revealed that Malaysia scored poorly in the provision of cancer survivorship care [10], with wide variations in access to basic services, such as psychosocial counselling and pain management services [11]. This is corroborated by country-specific findings from the ASEAN CosTs In ONcology (ACTION) study, which revealed high prevalence of Malaysian patients with impaired health-related quality of life and psychological distress at one year after their cancer diagnoses [12].

Needs assessment in cancer patients plays an integral role in facilitating the planning of effective cancer survivorship services. In the context of cancer survivorship care, several main constructs have been identified under the supportive care framework for cancer care, including psychological, physical, emotional, practical, informational, social, and spiritual needs [13]. Various instruments, such as the Supportive Care Needs Survey-Short Form (SCNS-SF) [14] and Cancer Survivors’ Unmet Needs Measure (CaSUN) [15], have been developed and are widely used to measure the needs of people living with cancer in different parts of the world. Nonetheless, variability in their methodological quality have been reported, including poor psychometric assessment of structural and criterion validity [16,17,18]. Although needs assessment tools, which are specific for women with breast cancer, have also been developed, such as the Breast Cancer Patient Needs Questionnaire (BR-CPNQ) [19], the Needs Self-rating Questionnaire for Breast Cancer (NSQ-BC) [20], and the Toronto Informational Needs Questionnaire-Breast Cancer (TINQ-BC) [21]; these tools are often focused on a specific construct of need or for patients in a specific phase of the cancer continuum. Although some of these tools have been culturally adapted and validated in several languages, they were largely developed in Western settings [18]. Prior evidence suggests cultural-specific differences in attitudes and prioritization of care between patient populations from these different settings [7]. To the best of our knowledge, no needs assessment tool has been developed for people living with cancer in the Southeast Asia region. There is, thus, a critical need for the development of a holistic needs assessment tool that is not only tailored to cater for the local health systems but is also sensitive to the regional socio-cultural practices. We, therefore, sought to develop and validate a dual language (English and Malay) needs assessment questionnaire for women with breast cancer in Malaysia.

## 2. Materials and Methods

The study involved two phases, namely the co-design phase (Phase I) and the psychometric testing phase (Phase II). Ethical approvals were obtained from the Medical Research and Ethics Committee (NMRR-18-3547-44456), University Malaya Medical Centre Medical Ethics Committee (2018108-6747), and Subang Jaya Medical Centre (201906.1).

### 2.1. Phase I: Co-Design of Questionnaire

The co-design approach was utilized to adequately capture the experiences and perspectives of Malaysian women living with breast cancer, healthcare professionals, and other local stakeholders, including cancer patient advocates, during the development of the questionnaire [22]. In this study, unmet need was defined as issues that patients experienced because of their cancer that they would require help or support with [18]. 

#### 2.1.1. Item Generation

Items were initially generated in English, guided by results from our recent qualitative studies that explored the needs of cancer patients in Malaysia [23,24,25,26,27]. A literature search of existing needs assessment tools for breast cancer patients was also conducted to identify any additional items that may be relevant for inclusion in the questionnaire. 

#### 2.1.2. Content and Face Validation

Content and face validation of the generated items were then carried out using a Delphi method with an eight-member expert panel comprising diverse stakeholders, including one representative from a breast cancer support group, one breast cancer survivor, one cancer nurse, one cancer counsellor, two breast surgeons, one clinical oncologist, and one psycho-oncologist. The expert panel was asked to provide a rating from 1 to 4 (1: not relevant, 2: not important, 3: relevant, 4: important) on the relevance of the generated items to the needs of breast cancer survivors, as well as on the comprehensiveness of the items. Ratings of 1 and 2 were considered “disagree”, while ratings of 3 and 4 were considered “agree”. The item content validity index (I-CVI) was calculated based on the proportion of the number of experts who agreed compared to the total number of experts for each item, while the scale content validity index (S-CVI) was determined by calculating the average of the I-CVIs. A cut-off value of ≥0.83 for the I-CVI and S-CVI was used [28]. 

Additionally, another five-member expert panel was asked to provide their recommendations on the measurement scale to be used for the questionnaire that they deemed to be most appropriate in regard to assessing patients’ unmet needs and what can be most clearly understood by patients. The Cohen’s Kappa of agreement was calculated in which a value of ≥0.60 was deemed acceptable [29]. 

#### 2.1.3. Translation 

While Malay is the official language of Malaysia, English is also widely used as a second language. As these two languages are often used interchangeably in Malaysia, we thus decided to provide a dual language questionnaire instead of individual languages to enable a composite understanding of the questionnaire items. The items were initially generated in English and were then translated from English to Malay by two bilingual members of the research team. The dual language items were then reviewed by breast cancer survivors who were proficient in both English and Malay. Their feedback was used to revise the translated items to improve item clarity and comprehensibility before cognitive pre-testing was conducted.

### 2.2. Phase II: Psychometric Testing of Questionnaire

#### 2.2.1. Study Population

Malaysian women above the age of 18 years who were diagnosed with breast cancer at least one-month prior to the time of recruitment were included. Patients with carcinoma in situ were excluded. Participants were recruited from a public academic hospital (University of Malaya Medical Center), two public Ministry of Health hospitals (Kuala Lumpur Hospital and National Cancer Institute), and a private hospital (Subang Jaya Medical Center) to ensure adequate representation of patients from diverse ethnic and socioeconomic backgrounds in Malaysia.

#### 2.2.2. Data Collection

Eligible participants were identified during their routine follow-ups at the respective study site and were briefed by their attending physicians or the study team. Those who agreed to participate were provided a participant information sheet and asked to complete an informed consent form. The questionnaires, which were available online on the Research Electronic Data Capture (REDCap) platform [30], were then administered face-to-face or via telephone call, by an interviewer, or self-administered depending on the individual patient’s preference. 

#### 2.2.3. Study Tools

All study participants were provided the dual language study questionnaire comprising items in both English and Malay languages. Participants were asked to rate the magnitude of their needs for each item over the past month (not applicable; no need/satisfied; low/a little; moderate/quite a bit; high/very much). Additionally, the European Organization for the Research and Treatment of Cancer Quality of Life Questionnaire (EORTC QLQ-C30) [31], which is widely used to assess health-related quality of life among people with cancer and has been validated previously in the local setting [32], was also administered for criterion validity analysis. Socio-demographic characteristics, along with medical details of the study participants were also collected. Clinical data were verified with medical records.

### 2.3. Statistical Analysis

Participant’s socio-demographic and clinical characteristics were described using frequencies and percentages. The study population were randomly divided into Sample 1 for EFA and Sample 2 for CFA. Construct validity was assessed using exploratory factor analysis (EFA) and confirmatory factor analysis (CFA) to determine convergent validity, divergent validity, and reliability of the questionnaire. The Kaiser–Meier–Olkin test (cut-off of ≥0.70) and Bartlett’s test of sphericity (*p* < 0.05) were first analyzed to determine sampling adequacy for factor analysis [33]. EFA was then conducted using parallel analysis based on the Horn’s approach using Promax rotation and principal component analysis through examination of eigenvalues to determine the optimal number of factors to be retained in the questionnaire [33]. This was followed by the CFA, which was conducted using the partial least square—structural equation modelling (PLS-SEM) approach [34]. Factor loadings and cross loadings of items were examined in which a cut-off of ≥0.30 for factor loadings was used in EFA while a cut-off of ≥0.50 was used in CFA to retain items [33,34]. To assess reliability, a cut-off value of ≥0.70 was used for both the Cronbach’s alpha and composite reliability, while a cut-off value of ≥0.50 was used to evaluate the average variance extracted [35,36]. Discriminant validity was established by examining the Heterotrait–Monotrait (HTMT) index, using a cut-off value of ≤0.85 [37]. Bifactor analysis was also conducted to determine the dimensionality of the study tool, whereby the percent of uncontaminated correlations (PUC) and explained common variance (ECV) were examined with cut-off values of >0.70, respectively [38]. Criterion validity was analyzed by measuring the correlation between the overall needs score and the global health score from the EORTC QLQ-C30 questionnaire using the Pearson’s correlation analysis. All analyses were conducted using FACTOR V12, IBM SPSS V22, IBM SPSS AMOS 28, bifactor indices calculator [39], and Smart PLS 4.

## 3. Results

### 3.1. Phase I: Co-Design of Questionnaire

#### 3.1.1. Item Generation and Content and Face Validation

A total of 67 items were initially generated. Following feedback from the expert panel, eight items were deleted, nine items were rephrased, and an additional seven items were added. The revised 66 items were then re-evaluated by the expert panel and the research team, following which 16 items were further dropped from the questionnaire. The item content validity index (I-CVI) and scale content validity index (S-CVI) for the remaining 50 items were adequate (Table 1). 

#### 3.1.2. Validation of Measurement Scale

Three measurement scales (scale A, B, and C) were initially drafted for the questionnaire (Table 2). Based on feedback from the expert panel, scale C was considered the most appropriate, with a CVI of 0.80 and Kappa agreement of 0.76. 

#### 3.1.3. Cognitive Pre-Test

Cognitive pre-testing of the questionnaire was conducted among 23 breast cancer survivors who were proficient in both languages. After completing the questionnaire, the participants were interviewed on the comprehensibility, relevancy, and acceptability of the items. Based on feedback by the participants, eleven items were reworded to improve clarity. Following this revision, all the items in the questionnaire were generally deemed as easily understandable, relevant, and acceptable by the participants. 

### 3.2. Phase II: Psychometric Testing of Questionnaire

#### 3.2.1. Participant Characteristics 

A total of 315 Malaysian women living with breast cancer were recruited in Phase 2 of the study. The majority were aged between 41 and 60 years (Table 3). Almost two-thirds of the participants were of Malay ethnicity (59.7%), followed by Chinese ethnicity at 26.0% and Indian ethnicity at 11.5%. Most participants had at least a secondary level education (50.5%), were retired (36.2%), and hailed from low-income households (68.9%). Approximately 30% owned either private or employer-sponsored health insurance. The majority of the women were recently diagnosed with breast cancer and had mostly stage 3 (29.2%) or metastatic disease (25.3%). 

#### 3.2.2. Exploratory Factor Analysis (EFA)

The Kaiser–Meier–Olkin measure of sampling adequacy was 0.88, while Bartlett’s test of sphericity was significant (*p* < 0.05). Parallel analysis using Horn’s approach extracted five factors. Item Q4 (I need healthcare professionals to be more compassionate during my treatment sessions and follow-ups) did not load on any factor and was thus dropped from the questionnaire. Items Q2 (I need access to counselling and psychological services as soon as possible after the doctor tells me that I have cancer) and Q5 (I need easy access to counselling and psychological services) cross loaded on factors 1 and 5. The research team decided to drop item Q2 as it was deemed similar to item Q5 and retained item Q5 in factor 1 due to its importance as an important need in the survivorship phase. Items Q3, Q9, Q20, and Q39 also cross-loaded into two factors but were retained in the factor with the highest factor loading due to their relevancy to cancer survivorship. After dropping item Q2, the eigenvalues of the factors ranged from 2.01 to 17.03 (Table 4). The factors were labelled as “emotional support” (containing 11 items), “social and intimate relationships” (6 items), “hospital appointments” (6 items), “personal care and health” (7 items), and “information and services” (18 items), which explained 17.03%, 4.15%, 2.89%, 2.09%, and 2.01% of the variances, respectively. The remaining 48 items had factor loadings above 0.30, which was considered adequate. The total variance explained by these five factors was 56.23%, which is above the recommended value of 50.0%, while the Cronbach alphas for each factor were excellent, ranging from 0.90 to 0.96, indicating good reliability of the questionnaire. 

#### 3.2.3. Confirmatory Factory Analysis (CFA)

In the CFA that utilized the PLS-SEM approach, the outer loadings of all 48 items were close to or greater than 0.50, which was considered adequate (Table 5, Figure 1). The composite reliability of each construct was greater than 0.80, while the average variance extracted was close to or exceeded 0.50, demonstrating good convergent validity and reliability among the five constructs. The Heterotrait–Monotrait (HTMT) ratios were below 0.85, indicating good discriminant validity between the constructs.

In the bifactor analysis, while the percent of uncontaminated correlations (PUC) at 0.77 exceeded the cut-off value of 0.70 for unidimensionality, the explained common variance (ECV) at 0.67 falls just below the cut-off. Although this indicates that the unidimensional model is a possibility, taken together with the results from the parallel analysis in the EFA, as well as from the CFA with the PLS-SEM model, the second-order model is the most optimal model for the NeAT-BC. 

#### 3.2.4. Criterion Validity

A significant negative correlation was observed between the total needs score and global health status (r = −0.14; *p* = 0.02). This indicated that higher scores for unmet needs were significantly correlated with lower scores for global health status.

#### 3.2.5. Respondent Burden 

Overall, the questionnaire took about 25 min to be completed and could be self-administered or interviewer-administered. Open-ended inquiries revealed that participants across all levels of education found the items in the questionnaire to be easily understandable and not overly intrusive or sensitive.

## 4. Discussion

Our validation exercise yielded a dual language needs assessment tool for breast cancer (NeAT-BC) that comprise 48 items, encompassing five domains, namely emotional support, social and intimate relationships, hospital appointments, and personal care and health, as well as information and services. Psychometric analyses showed that the dual language NeAT-BC possessed good reliability and validity. The instrument took approximately 25 min to be completed and could be either interviewer-administered or self-administered.

Prior studies have reported high prevalence of psychological distress among Malaysian women with breast cancer at the time of cancer diagnosis, which persisted at 12 months after cancer diagnosis [12,40]. Our recent qualitative studies among Malaysians living with cancer underscored major gaps in accessing cancer support groups, as well as in experiences with discrimination and stigma due to their cancer diagnosis [26]. In addition to emotional distress, unmet financial needs, such as those related to health insurance reimbursements, were also highlighted [23,27]. Considering the importance of these survivorship issues in the local setting, the items related to these matters were retained in the questionnaire during factor analysis despite presence of cross-loadings. 

Our finding of an inverse correlation between overall needs and quality of life status is consistent with prior literature [41,42] and was also reported in the validation of the comprehensive needs assessment tool [43]. As described by Shim et al. (2011) [43], the weak correlation could be because our comprehensive questionnaire included items that directly impacted wellbeing, such as those related to emotional support and social and intimate relationships, as well as items that indirectly impacted wellbeing, such as items involving hospital appointments, as well as information and services constructs. Nonetheless, the negative association between scores from NeAT-BC and global health status indicate that our tool possesses good criterion validity and can be used to predict quality of life among women living with breast cancer.

We had attempted to ensure that the developed questionnaire included items that were comprehensive but at the same time was of feasible length to reduce respondent burden, which can be influenced by several factors, including the length of the questionnaire and mode of questionnaire administration [44]. Compared to other needs assessment tools [18], which ranged between 9 to 138 items and had completion time of between 5 and 76 min, our 48-item questionnaire seems to contribute to a low respondent burden as it took about 25 min to be completed and was deemed easily understandable, acceptable, and non-intrusive by the participants. Additionally, the NeAT-BC may not only be interviewer-administered, but also be self-administered, thus reducing the administrative burden on the healthcare system. 

Compared to other breast-cancer-specific needs questionnaires, such as the Self-assessed Support Needs-Breast Cancer (SASN-BR) [45], and Breast Cancer Patients’ Need’ Questionnaire (BR-CNPQ) [19], the newly developed NeAT-BC contained a larger number of items related to financial and employment needs. The above observation may be explained by differences in the time period when the earlier tools were developed in, i.e., in early 2000s, where there were far fewer designer drugs for breast cancer, and many of the anticancer therapies were relatively more toxic compared to that of today [46]. Notably, targeted therapies for breast cancer, which are costly, were not available until more recently in many parts of the world, including in Malaysia [47]. Thus, compared to financial- and employment-related needs, dealing with side effects of treatments, as well as psychological-related issues may have served as more pressing needs for women living with breast cancer about two decades ago. This is supported by the contents of needs assessment tools that were more recently developed, such as the Breast Cancer Survivors Rehabilitation Care Needs Questionnaire that was developed in 2016 in Belgium [48] and the Short Form Survivor Unmet Needs Survey (SF-SUNS) developed in 2014 in Canada [49], in which financial and employment related needs were the main domains. The NeAT-BC also contained two items related to traditional and complementary medicine (TCM), reflecting the local socio-cultural beliefs and practice of using these therapies. Intriguingly, to the best of our knowledge, items on TCM-related unmet needs were present in the Cancer Survivors Unmet Needs (CaSUN) and SF-SUNS tools [15,49], both of which were developed in higher-income Western settings, but not in recently developed tools from Asia, such as the Sri Lankan Informational Needs Questionnaire for Breast Cancer from Sri Lanka [50] and the Patient Needs Questionnaire from China [51]. 

In the clinical setting, the NeAT-BC may aid the oncology team or patient navigators to identify patients with unmet needs and direct them to the providers of the needed supportive care. The tool may also be used for research purposes by the healthcare system and non-governmental organizations to identify gaps in provision of supportive services, strengthen existing services or plan new services to meet areas with unmet needs. Although the NeAT-BC provides an avenue for the patients’ voices to be heard in healthcare, we would like to emphasize that the tool by itself will not be able to address patients’ unmet needs. To this end, the translation of findings from needs assessment into effective actions requires strong coordination and partnership between different health disciplines and external stakeholders towards the provision of a clear supportive care pathway [52].

### Study Strengths and Limitations

A major strength of the study lies in the use of data from previous qualitative studies among cancer survivors in the country, as well as the co-design approach to guide item generation in the development phase of the NeAT-BC. This increases the confidence that needs that are relevant and important to women living with breast cancer in the local setting are adequately represented and included in the questionnaire. Furthermore, unlike many existing needs assessment questionnaires, which focus only on a specific period of cancer survivorship, the NeAT-BC can be administered to breast cancer survivors at all phases of the cancer survivorship period. Nonetheless, as the majority of our study respondents were diagnosed with breast cancer within the past 6 months at time of recruitment, future studies should be conducted with a larger patient sample of women who were diagnosed more than 6 months ago to further validate the utility of the NeAT-BC among longer-term cancer survivors. The dimensionality of the NeAT-BC should also be further explored in future studies with a larger patient sample. To the best of our knowledge, however, our newly developed questionnaire is the only needs assessment tool that has been developed in Southeast Asia. Despite certain differences in the health systems and socio-cultural practices between the different countries in Southeast Asia, the NeAT-BC can still serve as an important tool to be adapted for use in these regions due to the similar broader socio-cultural beliefs. However, due to the lack of a gold standard for comprehensive needs assessment among Asian breast cancer survivors, the criterion validity of NeAT-BC could not be directly measured. For patients’ convenience, telephone calls were also used as a mode of survey administration, which we acknowledge could present response bias. 

## 5. Conclusions

The dual language needs assessment tool for breast cancer (NeAT-BC) demonstrated good reliability and validity, with low respondent burden. The utilization of the validated NeAT-BC to determine the type and magnitude of unmet needs following breast cancer may be useful in guiding the establishment of evidence-informed cancer survivorship services in Malaysia. The tool may also have wider clinical and research utility in other multi-ethnic settings, as well as within the broader Southeast Asia region. 

## Figures and Tables

**Figure 1 diagnostics-13-00241-f001:**
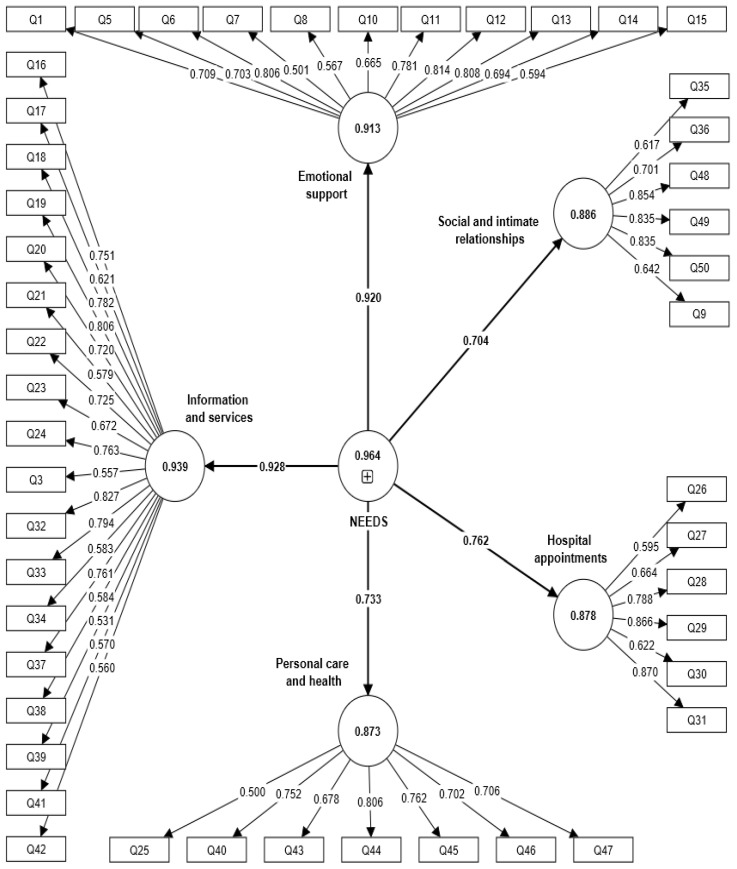
Confirmatory factory analysis (CFA) model of the needs assessment tool for breast cancer (NeAT-BC) using the partial least square—structural equation modelling (PLS-SEM) approach.

**Table 1 diagnostics-13-00241-t001:** Content validation of the needs assessment tool for breast cancer (NeAT-BC).

Item Number	Item	I-CVI
Q1	I need healthcare professionals to be understanding and sensitive to my feelings when breaking the news that I have cancer Saya memerlukan anggota kesihatan agar memahami dan prihatin terhadap perasaan saya ketika menerima keputusan diagnosis kanser	1.00
Q2	I need access to counselling and psychological services as soon as possible after the doctor tells me that I have cancer Saya memerlukan akses kepada perkhidmatan kauseling dan psikologi selepas doktor memberitahu bahawa saya mempunyai kanser dengan secepat mungkin	1.00
Q3	I need to be informed about cancer support groups as soon as possible Saya perlu diberitahu mengenai kumpulan sokongan kanser dengan secepat mungkin	1.00
Q4	I need healthcare professionals to be more compassionate during my treatment sessions and follow-ups Saya memerlukan anggota kesihatan agar lebih bersimpati terhadap saya semasa menjalani rawatan dan rawatan susulan	1.00
Q5	I need easy access to counselling and psychological services Saya memerlukan akses yang mudah kepada perkhidmatan kaunseling dan psikologi	0.88
Q6	I need religious or spiritual support to cope with my emotions Saya memerlukan sokongan agama atau rohani untuk mengatasi masalah emosi saya	1.00
Q7	I need help to cope with my worries and fear of recurrence Saya memerlukan bantuan untuk mengatasi kebimbangan dan ketakutan yang kanser saya bakal berulang	1.00
Q8	I need help to accept changes in my body (e.g.,: surgery scars, loss of breast) and to boost my body image and self confidence Saya memerlukan pertolongan untuk menerima perubahan pada tubuh badan saya (misalnya: parut pembedahan, kehilangan payudara) dan untuk meningkatkan imej badan dan keyakinan diri	0.88
Q9	I need to deal with uncomfortable situations in social setting because of my cancer (e.g., discrimination, stigma, stereotype) Saya perlu menangani situasi yang kurang menyenangkan dalam persekitaran sosial kerana menghidap kanser [cth: diskriminasi, stigma, tanggapan yang salah]	0.88
Q10	I need to join a cancer support group Saya perlu menyertai kumpulan sokongan kanser	1.00
Q11	I need my family to give me more emotional support Saya memerlukan keluarga saya untuk memberi lebih banyak sokongan emosi/moral kepada saya	0.88
Q12	I need my close family members to receive emotional support Saya memerlukan ahli keluarga terdekat saya untuk menerima sokongan emosi/moral	1.00
Q13	I need my spouse/family to be given information regarding my cancer Saya memerlukan pasangan/keluarga saya untuk diberi maklumat mengenai kanser saya	1.00
Q14	I need help to care for my family members who are depending on me (e.g., children, parents, spouses) Saya memerlukan bantuan untuk menjaga ahli keluarga yang bergantung pada saya [cth: kanak-kanak, ibu-bapa, pasangan]	0.75
Q15	I need help to carry out my daily activities, such as cooking, childcare, cleaning, etc. Saya memerlukan pertolongan bagi menjalani kegiatan seharian seperti memasak, menjaga anak, membersihkan rumah dll	0.88
Q16	I need the information that the doctor gives me to be easily understood Saya memerlukan doktor untuk memberi maklumat kesihatan yang mudah difahami	1.00
Q17	I need the doctor to give me written information about my cancer to read later on Saya memerlukan doktor untuk memberi saya maklumat bertulis mengenai kanser saya untuk dibaca kemudian	0.88
Q18	I need adequate information about all the different treatment options before I choose to have them Saya perlu maklumat yang secukupnya mengenai kesemua pilihan, rawatan yang berbeza sebelum saya memilih untuk menjalaninya	0.88
Q19	I need to be informed about my test results as soon as they are known Saya perlu dimaklumkan mengenai keputusan ujian saya sebaik sahaja ia diketahui	0.88
Q20	I need more cancer-related materials (books, brochures, videos, etc.) in the waiting area or online at the hospital website Saya memerlukan lebih banyak bahan berkaitan kanser [buku, risalah, video, dll] di ruang menunggu atau dalam talian di laman web hospital	1.00
Q21	I need my oncologist to openly discuss traditional and complementary medicine with me Saya memerlukan doktor kanser saya untuk membincangkan secara terbuka mengenai perubatan tradisional dan perubatan komplementari dengan saya	1.00
Q22	I need information on things that I can do to take better care of myself Saya memerlukan maklumat mengenai tindakan yang boleh saya ambil untuk menjaga diri saya dengan lebih baik	0.88
Q23	I need information about diet to know what I should avoid or what should I be eating more Saya memerlukan maklumat pemakanan untuk mengetahui tentang makanan yang harus saya elakkan atau yang harus saya makan dengan lebih banyak	1.00
Q24	I need my doctor to manage better the side effects of my cancer and treatments Saya memerlukan doktor saya untuk menguruskan kesan sampingan kanser dan rawatan saya dengan sebaik mungkin	1.00
Q25	I need my family doctor/GP to be also knowledgeable about the side effects of my cancer treatment Saya memerlukan doktor keluarga /doktor am saya juga untuk berpengetahuan tentang kesan sampingan rawatan kanser saya	0.88
Q26	I need to be seen by the same team of doctors at every appointment. Saya perlu dilihat oleh doktor-doktor dari pasukan rawatan yang sama pada setiap temujanji	0.88
Q27	I need the waiting time in the hospital to be shortened Saya memerlukan masa menuggu di hospital untuk dipendekkan	0.88
Q28	I need to be given an explanation if there is going to be a delay in seeing the doctor Saya perlu diberi penjelasan jika terdapat kelewatan untuk berjumpa dengan doktor	1.00
Q29	I need help in making appointments and someone to call if I need to change appointments Saya memerlukan pertolongan dalam membuat temujanji dan seseorang untuk menelefon jika saya perlu menukar tarikh temujanji	1.00
Q30	I need all my hospital appointments to be set on the same day whenever possible Saya memerlukan semua temujanji hospital saya untuk ditetapkan pada hari yang sama sekiranya boleh	0.88
Q31	I need to know who to contact if I have any questions or concerns on my disease or treatment in between hospital appointments Saya perlu tahu siapa yang patut dihubungi sekiranya saya mempunyai sebarang kemuskilan mengenai penyakit atau rawatan saya di antara temujanji	1.00
Q32	I need the hospital facilities and surroundings to be clean, comfortable and pleasant Saya memerlukan kemudahan dan persekitaran hospital yang bersih, selesa dan menyenangkan	1.00
Q33	I need the hospital facilities to be located near each other (e.g., pharmacy, clinic, payment counter, wards) Saya memerlukan kemudahan hospital terletak berhampiran antara satu sama lain [misalnya, farmasi, klinik, kaunter pembayaran, wad]	1.00
Q34	I need the hospital to provide traditional and complementary medicine services along with normal hospital treatment Saya memerlukan pihak hospital untuk menawarkan perkhidmatan perubatan tradisional dan komplementari di samping perubatan biasa di hospital	0.88
Q35	I need information and help in coping with my sexual difficulties Saya memerlukan maklumat dan perkhidmatan bagi mengatasi masalah seksual	0.88
Q36	I need to be informed about fertility issues, and potential solutions before my treatment Saya perlu dimaklumkan mengenai isu-isu kesuburan serta alternatif penyelesaiannya, sebelum menjalani rawatan kanser	0.75
Q37	I need information on the costs of my treatments before starting them Saya memerlukan maklumat tentang kos rawatan saya sebelum memulakan rawatan	0.88
Q38	I need assistance to pay for my cancer treatments Saya memerlukan bantuan untuk membayar rawatan kanser saya	0.75
Q39	I need help to understand my insurance benefits and coverage, and in making claims Saya memerlukan bantuan untuk memahami manfaat dan liputan insurans saya, serta untuk membuat tuntutan	0.88
Q40	I need to buy health insurance after my cancer diagnosis Saya perlu membeli insurans kesihatan selepas menerima diagnosis kanser	0.88
Q41	I need guidance and assistance in obtaining financial assistance Saya memerlukan bimbingan, dan pertolongan untuk mendapat bantuan kewangan	0.75
Q42	I need affordable parking and transportation when I come to the hospital Saya memerlukan tempat letak kereta dan pengangkutan pada harga yang berpatutan apabila ke hospital	1.00
Q43	I need affordable special mastectomy bras, prosthesis or wigs Saya memerlukan coli khas, payudara palsu atau rambut palsu pada harga yang berpatutan	0.88
Q44	I need help to pay for dietary supplements (example: special milk, special food) Saya memerlukan bantuan untuk membayar makanan tambahan [contoh: susu khas, makanan istimewa]	0.88
Q45	I need affordable special equipments (e.g., wheelchair, special bed) Saya memerlukan peralatan perubatan khas pada harga yang berpatutan [cth: kerusi roda, katil khas]	0.88
Q46	I need to pay for hired help at home (e.g., maid, babysitter) Saya perlu mengupah seseorang bagi membantu menjalankan kerja harian di rumah [cth: pembantu rumah, pengasuh]	0.88
Q47	I need help to cope with a reduced household income due to my illness Saya memerlukan pertolongan untuk menangani pendapatan isi rumah yang berkurangan disebabkan oleh penyakit saya	0.88
Q48	I need to be treated fairly at my workplace despite having cancer Saya perlu dilayan dengan adil di tempat kerja walaupun menghidap kanser	0.88
Q49	I need workplace flexibility (e.g., time off for hospital appointments, changes in job scope, flexible hours, work from home) Saya memerlukan pengubahsuaian di tempat kerja [cth: masa khas untuk menghadiri temujanji hospital, perubahan tugas-tugas kerja]	1.00
Q50	I need help to find a new job after my cancer diagnosis Saya memerlukan bantuan untuk mencari pekerjaan baharu selepas didiagnosa dengan kanser	0.88
S-CVI		0.92

I-CVI ≥ item content validity index; S-CVI = scale content validity index.

**Table 2 diagnostics-13-00241-t002:** Scale validation of the needs assessment tool for breast cancer (NeAT-BC).

Scale		CVI	Kappa
A	No needs, or not applicable; Have need but need is being met; Weak; Moderate; Strong.	0.20	0.05
B	No need/not applicable; No need/satisfied; Rarely; Sometimes; Always.	0.40	0.13
C	Not applicable; No need/satisfied; Low/a little; Moderate/quite a bit; High/Very much.	0.80	0.76

CVI = content validity index.

**Table 3 diagnostics-13-00241-t003:** Sociodemographic characteristics of study participants (n = 315).

Characteristics	Overall n = 315 n (%)	Sample 1 n = 152 n (%)	Sample 2 n = 163 n (%)
**Age (years)**			
Mean (SD)	53 (10)	53 (10)	53 (11)
≤40	41 (13.2)	20 (13.2)	21 (13.1)
41–60	196 (63.0)	103 (68.2)	93 (58.1)
≥61	74 (23.8)	28 (18.5)	46 (28.7)
Unknown	4	1	3
**Ethnicity**			
Malay	188 (59.7)	90 (59.2)	98 (60.1)
Chinese	82 (26.0)	44 (28.9)	38 (23.3)
Indian	37 (11.7)	16 (10.5)	21 (12.9)
Others	8 (2.5)	2 (1.3)	6 (3.7)
**Marital status**			
Single	37 (11.7)	19 (12.5)	18 (11.0)
Married	238 (75.6)	115 (75.7)	123 (75.5)
Others	40 (12.7)	18 (11.8)	22 (13.5)
**Highest education level attained**			
Primary and below	29 (9.2)	10 (6.6)	19 (11.7)
Secondary	159 (50.5)	76 (50.0)	83 (50.9)
Tertiary	127 (40.3)	66 (43.4)	61 (37.4)
**Employment status**			
Employed	99 (31.4)	53 (34.9)	46 (28.2)
Self-employed	11 (3.5)	6 (3.9)	5 (3.1)
Housewife	48 (15.2)	17 (11.2)	31 (19.0)
Retired	114 (36.2)	55 (36.2)	59 (36.2)
Unemployed	43 (13.7)	21 (13.8)	22 (13.5)
**Household income**			
MYR ≤4360	217 (68.9)	106 (69.7)	111 (68.1)
MYR 4361-MYR 6919	62 (19.7)	29 (19.1)	33 (20.2)
MYR ≥6920	36 (11.4)	17 (11.2)	19 (11.7)
**Ownership of health insurance**			
None	220 (71.2)	102 (68.0)	118 (74.2)
Yes	89 (28.8)	48 (32.0)	41 (25.8)
Unknown	6	2	4
**Time since diagnosis**			
≤6 months	246 (78.1)	123 (80.9)	123 (75.5)
>6 months	69 (21.9)	29 (19.1)	40 (24.5)
**Cancer stage at initial diagnosis**			
I	40 (12.8)	17 (11.3)	23 (14.2)
II	102 (32.7)	54 (36.0)	48 (29.6)
III	91 (29.2)	43 (28.7)	48 (29.6)
IV	79 (25.3)	36 (24.0)	43 (26.5)
Unknown	3	2	1

Sample 1: study population included in exploratory factor analysis; Sample 2: study population included in confirmatory factor analysis.

**Table 4 diagnostics-13-00241-t004:** Exploratory factor analysis of the needs assessment tool for breast cancer (NeAT-BC).

Item	Factors				
	1	2	3	4	5
	Emotional Support	Social and Intimate Relationships	Hospital Appointments	Personal Care and Health	Information and Services
Q1	0.31				
Q5	0.48				
Q6	0.52				
Q7	0.58				
Q8	0.47				
Q10	0.52				
Q11	0.64				
Q12	0.71				
Q13	0.72				
Q14	0.68				
Q15	0.92				
Q35		0.72			
Q36		0.84			
Q48		0.63			
Q49		0.67			
Q50		0.65			
Q9		0.37			
Q26			0.71		
Q27			0.64		
Q28			0.55		
Q29			0.64		
Q30			0.89		
Q31			0.71		
Q25				0.51	
Q40				0.66	
Q43				0.68	
Q44				0.65	
Q45				0.78	
Q46				0.72	
Q47				0.62	
Q3					0.46
Q16					0.79
Q17					0.74
Q18					0.86
Q19					0.84
Q20					0.36
Q21					0.73
Q22					0.52
Q23					0.62
Q24					0.77
Q32					0.80
Q33					0.51
Q34					0.66
Q37					0.66
Q38					0.51
Q39					0.54
Q41					0.68
Q42					0.50
Eigenvalues	17.03	4.15	2.89	2.09	2.01
% of variance	32.06	8.30	5.78	4.17	4.02
Cronbach’s alpha	0.93	0.90	0.90	0.91	0.96

**Table 5 diagnostics-13-00241-t005:** Confirmatory factor analysis of the needs assessment tool for breast cancer (NeAT-BC) using PLS-SEM approach.

Construct	Items	Outer Loading	Composite Reliability	Average Variance Extracted
C1 Emotional support	Q7	0.50	0.91	0.49
	Q8	0.57		
	Q15	0.59		
	Q10	0.67		
	Q14	0.69		
	Q5	0.70		
	Q1	0.71		
	Q11	0.78		
	Q6	0.81		
	Q13	0.81		
	Q12	0.81		
C2 Social and intimate relationships	Q35	0.62	0.89	0.57
	Q9	0.64		
	Q36	0.70		
	Q49	0.84		
	Q50	0.84		
	Q48	0.85		
C3 Hospital appointments	Q26	0.60	0.88	0.55
	Q30	0.62		
	Q27	0.66		
	Q28	0.79		
	Q29	0.87		
	Q31	0.87		
C4 Personal care and health	Q25	0.50	0.87	0.50
	Q43	0.68		
	Q46	0.70		
	Q47	0.71		
	Q40	0.75		
	Q45	0.76		
	Q44	0.81		
C5 Information and services	Q39	0.53	0.94	0.47
	Q3	0.56		
	Q42	0.56		
	Q41	0.57		
	Q21	0.58		
	Q34	0.58		
	Q38	0.58		
	Q17	0.62		
	Q23	0.67		
	Q20	0.72		
	Q22	0.73		
	Q16	0.75		
	Q37	0.76		
	Q24	0.76		
	Q18	0.78		
	Q33	0.79		
	Q19	0.81		
	Q32	0.83		

PLS-SEM = partial least square—structural equation modelling.

## Data Availability

Data are available from the study’s principal investigator upon reasonable request.

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
