# Peer review of "Development and Validation of a Dual-Language (English and Malay) Needs Assessment Tool for Breast Cancer (NeAT-BC)"

_diagnostics, 2023, doi:10.3390/diagnostics13020241_

Round 1

Reviewer 1 Report

Thank you for giving me the opportunity to review this manuscript entitled “Development and Validation of a Dual Language (English and Malay) Needs Assessment Tool for Breast Cancer (NeAT-BC)”. This manuscript has some interesting findings and benefits the readers. My comments are listed and I hope that the authors will be willing to address them.

Introduction:

1.    The focus of this study is on the development and psychometric evaluation of a need assessment tool. Yet only a small portion of the content is focused on this topic. To enhance the balance and the flow of story, the authors are suggested to include more content on discussing literature and what has been known regarding the status of needs assessment. For instance, what are the psychometric issues that exist in these assessments? What are the existing needs assessments that have been used in the cancer population? Any issues and problems that can be improved?

2.    The authors should define needs assessment clearly since this is the most important element throughout the manuscript. What is the construct of needs? How many different needs are there in the context of cancer rehabilitation? Which needs were being studied in this paper (selective or inclusive?).

3.    The authors should clarify the differences between western and Asian settings that make the existing tools not applicable to Asian population. Doing so will strengthen the importance of this study.

Methods and Results:

1.    How was the optimal number of factors in EFA determined; clarification about the criteria is needed. It will be informative to see the scree plots as well.

2.    Have different structure (factor model) of 5-factor be examined with CFA (e.g., bifactor, second-order, first-order etc)? Also, it would be helpful for the readers to view it the final model graphically so that the structure, factor loading, factor correlation, and correlated errors can be immediately comprehended.

3.    Adding abbreviations to each item number in the table will be helpful. The current tables with different numerical values (Q1, Q2 ….) are difficult for understanding what they are about.

Discussion:

1.    A paragraph that is missing is a discussion on the comparison of findings or final scale from this study with other needs assessment for other clinical population (cancer/chronic disease) and the western cohort. It would be meaningful to discuss the commonalities and differences, as they reveal important information regarding breast cancer survivors in Malaysia. I believe such discussion will lead to clinician implications that clinicians working in Malaysia may benefit from.

2.    Time since diagnosis is another important factor that should be considered when interpreting the findings of the study. The majority of sample were <6 months; cancer survivors at a more chronic phase experience more changes in their lives in home, work, and community and may lead to different findings.

3.    In this regard, it is important to further highlight the impact of socio-demographic status (e.g., income) of the study population on needs as well as the findings.

Author Response

We appreciate the reviewer's comments and thank the reviewer for taking the time to review our manuscript. Our response to the comments are as attached. 

Reviewer 2 Report

3.1.2 REVIEW: “Development and Validation of a Dual Language (English and Malay) Needs Assessment Tool for Breast Cancer (NeAT-BC)

The authors set out to “develop and validate a dual language needs assessment tool for breast cancer”. In general, I think that they have achieved this. In particular, I agree with their concluding statement that “A major strength of the study lies in the use of data from previous qualitative studies among cancer survivors in the country, as well as the co-design approach, to guide item generation in the development phase of the NeAT-BC”. The authors should be congratulated on a well conceptualized study, a well written manuscript and very competent data analysis. While well-written there are occasions where the authors could have and should have provided more detail.

While classical test theory is a popular and accepted way for examining the psychometric properties of an instrument, my own view is that in the development of an instrument, a combination of classical test theory and item response theory are more applicable. However, this observation does not detract from the quality and contribution of the manuscript.

My notes are below

-        Abstract, line 18: should it not be for “women with breast cancer” rather than “breast cancer”?

-        Line 40: I think it should be “awareness of”, not “awareness on”

-        Lines 108-109: please provide more information on the process in establishing I-CVI and S-CVI. What was required of the expert panel in terms of rating the items? How were items scored? And how were the two indices calculated?

-        Was there no back-translation?

-        Line 134: “to fill” should perhaps be “to complete”

-        It is mentioned that all participants were provided with the dual language study questionnaire. This is not very clear. Were all participants provided with both the English and Malay version of the questionnaire and what was the purpose of that?

-        The EORTC QLQ-C30 requires a fuller description. What does it assess? What are its’ psychometric properties?

-        Lines 161-162: a cut-off value cannot be used to “determine” AVE. It should probably be to “assess” or “evaluate”.

-        I do not think that paragraph 3.1.2 is about “scale validation”. As far as I can see it seems to deal with the response format of the questionnaire.

-        Line 195 and 199: The majority

-        Could the mean age of the participants be provided?

-        The sentence in table 3 header relating to the division of the sample into two groups should rather be placed in the analysis section.

-        Line 214: “into any factor” should be “on any factor”.

-        EFA: what extraction and rotation method were used?

-        Table 4: it is not clear whether these are the factor loadings of the items after item 2 was dropped? Was EFA conducted again after item 2 was dropped? It would appear that the eigenvalues at the bottom of Table 4 are the same as those reported on page 9 prior to dropping item 2. The author should be re-run the factor analysis as deleting an item will lead to different eigenvalues as well as different factor loadings.

-        Line 232: “close to or exceeded 0.50” as some AVE’s was above 0.50

-        What model was tested with CFA (i.e., correlated five-factor, bi-factor, second-order factor etc.). Was it compared to any other model (e.g., one-factor)?

-        There were also no fit indices reported.

-        Minor note: the symbol on line 240 is a dash (-) and not a minus ().

Thank you for the opportunity to provide input. I hope the authors will find it useful and constructive.

Regards

Author Response

The authors set out to “develop and validate a dual language needs assessment tool for breast cancer”. In general, I think that they have achieved this. In particular, I agree with their concluding statement that “A major strength of the study lies in the use of data from previous qualitative studies among cancer survivors in the country, as well as the co-design approach, to guide item generation in the development phase of the NeAT-BC”. The authors should be
congratulated on a well conceptualized study, a well written manuscript and very competent data analysis. While well-written there are occasions where the authors could have and should have provided more detail.

While classical test theory is a popular and accepted way for examining the psychometric properties of an instrument, my own view is that in the development of an instrument, a combination of classical test theory and item response theory are more applicable. However, this observation does not detract from the quality and contribution of the manuscript.

My notes are below
- Abstract, line 18: should it not be for “women with breast cancer” rather than “breast cancer”?

We have amended as suggested in the abstract.

- Line 40: I think it should be “awareness of”, not “awareness on”

We have amended as suggested in the introduction section on page 1, paragraph 1.

“Improvement in survival rates and the resulting growing population of breast cancer survivors have brought about an increased awareness of the importance of addressing their supportive care needs that goes beyond provision of cancer therapies alone [1]. …”

- Lines 108-109: please provide more information on the process in establishing I-CVI and SCVI. What was required of the expert panel in terms of rating the items? How were items scored? And how were the two indices calculated?

We have added an explanation as suggested in the Materials and Methods section on page 3, paragraph 4.

“… The expert panel was asked to provide a rating from 1 to 4 (1: not relevant, 2: not important, 3: relevant, 4: important) on the relevance of the generated items to the needs of breast cancer survivors, as well as on the comprehensiveness of the items. Ratings of 1 and 2 were considered “disagree” while ratings of 3 and 4 were considered “agree”. The item content validity index (I-CVI) was calculated based on the proportion of the number of experts who agreed over the total number of experts for each item, while the scale content validity index (SCVI) was determined by calculating the average of the I-CVIs. A cut-off value of ≥ 0.83 for the I-CVI and S-CVI was used [28].”

- Was there no back-translation?

We did not perform a back-translation. Instead, both the English and Malay versions of the items were reviewed by breast cancer survivors who were proficient in both languages to ensure item clarity, comprehensibility and consistency.

- Line 134: “to fill” should perhaps be “to complete”

We have revised as suggested on page 4, paragraph 2.

 “… Those who agreed to participate were provided a participant information sheet and asked to complete an informed consent form. …”

- It is mentioned that all participants were provided with the dual language study questionnaire. This is not very clear. Were all participants provided with both the English and Malay version of the questionnaire and what was the purpose of that?

Yes, all participants were provided the dual language questionnaire that comprised both the English and Malay versions of the items. While Malay is the official national language in Malaysia, English is also widely used. Given that these two languages are often used interchangeably in Malaysia, we thus decided to provide a dual language questionnaire instead of by individual languages.

We have specified this explanation on page 3, paragraph 6, and also on page 4, paragraph 3.

“While Malay is the official language of Malaysia, English is also widely used as a second language. As these two languages are often used interchangeably in Malaysia, we thus decided to provide a dual language questionnaire instead of by individual languages to enable a composite understanding of the questionnaire items. …”

“All study participants were provided the dual language study questionnaire comprising items in both English and Malay languages. …”

- The EORTC QLQ-C30 requires a fuller description. What does it assess? What are its’ psychometric properties?

We have added as suggested on page 4, paragraph 3.

“… Additionally, the European Organization for the Research and Treatment of Cancer Quality of Life Questionnaire (EORTC QLQ-C30) [31], which is widely used to assess health-related quality of life among people with cancer and have been validated previously in the local setting [32], was also administered for criterion validity analysis. …”

- Lines 161-162: a cut-off value cannot be used to “determine” AVE. It should probably be to “assess” or “evaluate”.

We have amended as suggested on page 4, paragraph 4.

“… To assess reliability, a cut-off value of ≥0.70 was used for both the Cronbach’s alpha and composite reliability, while a cut-off value of ≥0.50 was used to evaluate the average variance extracted [35-36]. …”

- I do not think that paragraph 3.1.2 is about “scale validation”. As far as I can see it seems to deal with the response format of the questionnaire.

We have now amended the heading to “Validation of measurement scale” (see page 7).

- Line 195 and 199: The majority

We have amended as suggested on page 8, paragraph 2.

“… The majority were aged between 41 and 60 years (Table 3). … The majority of the women were recently diagnosed with breast cancer and had mostly stage 3 (29.2%) or metastatic disease (25.3%).”

- Could the mean age of the participants be provided?

We have provided the mean age as suggested in Table 3 (page 9).

- The sentence in table 3 header relating to the division of the sample into two groups should rather be placed in the analysis section.

We have removed the sentence from Table 3 and have added in into the statistical analysis subsection as suggested on page 4, paragraph 4.

“… The study population were randomly divided into Sample 1 for EFA and Sample 2 for CFA.…”

- Line 214: “into any factor” should be “on any factor”.

We have amended as suggested on page 10, paragraph 1.

“… Item Q4 (I need healthcare professionals to be more compassionate during my treatment sessions and follow-ups) did not load on any factor and was thus dropped from the questionnaire. …”

- EFA: what extraction and rotation method were used?

The extraction and rotation method are as specified on page 4, paragraph 4.

“… EFA was then conducted using parallel analysis based on the Horn’s approach using Promax rotation and principal component analysis through examination of eigenvalues to determine the optimal number of factors to be retained in the questionnaire [33]. …”

- Table 4: it is not clear whether these are the factor loadings of the items after item 2 was dropped? Was EFA conducted again after item 2 was dropped? It would appear that the eigenvalues at the bottom of Table 4 are the same as those reported on page 9 prior to dropping item 2. The author should be re-run the factor analysis as deleting an item will lead to different eigenvalues as well as different factor loadings.

The eigenvalues and factor loadings reported on page 9 and in Table 4 are from the EFA after item 2 has been dropped. We have revised this in the manuscript on page 10, paragraph 1 to clarify this.

“… Parallel analysis using Horn’s approach extracted five factors. Item Q4 (I need healthcare professionals to be more compassionate during my treatment sessions and follow-ups) did not load on any factor and was thus dropped from the questionnaire. Items Q2 (I need access to counselling and psychological services as soon as possible after the doctor tells me that I have cancer) and Q5 (I need easy access to counselling and psychological services) cross loaded on factors 1 and 5. The research team decided to drop item Q2 as it was deemed similar to item Q5 and retained item Q5 in factor 1 due to its importance as an important need in the survivorship phase. Items Q3, Q9, Q20, and Q39 also cross-loaded into two factors but were retained in the factor with the highest factor loading due to their relevancy to cancer survivorship. After dropping item Q2, the eigenvalues of the factors ranged from 2.01 to 17.03 (Table 4). The factors were labelled as “emotional support” (containing 11 items), “social and intimate relationships” (6 items), “hospital appointments” (6 items), “personal care and health” (7 items), and “information and services” (18 items), which explained 17.03%, 4.15%, 2.89%, 2.09%, and 2.01% of the variances respectively. The remaining 48 items had factor loadings above 0.30, which was considered adequate. The total variance explained by these five factors was 56.23%, which is above the recommended value of 50.0%, while the Cronbach alphas for each factor were excellent, ranging from 0.90 to 0.96, indicating good reliability of the questionnaire.”

- Line 232: “close to or exceeded 0.50” as some AVE’s was above 0.50

We have amended as suggested on page 12, paragraph 1.

“… …, while the average variance extracted was close to or exceeded 0.50, …”

- What model was tested with CFA (i.e., correlated five-factor, bi-factor, second-order factor etc.). Was it compared to any other model (e.g., one-factor)?

The CFA was performed based on results of the EFA in another set of data from the same study population (Sample 1 for EFA and Sample 2 for CFA). We ran the second-order model using the SMART-PLS software. We have now added the final CFA model as shown in Figure 1 on page 13.

- There were also no fit indices reported.

Since we used the SMART PLS software for CFA, there were no criterias for fitting indices. Instead, the outer loading of all items, composite reliability of each construct, and the average variance extracted were reported to indicate convergent validity and reliability. We also reported the Heterotrait-Monotrait (HTMT) ratios to demonstrate discriminant validity. These are reported on page 11, paragraph 1.

“In the CFA that utilized the PLS-SEM approach, the outer loadings of all 48 items were close to or greater than 0.50, which was considered adequate (Table 5, Figure 1). The composite reliability of each construct was greater than 0.80, while the average variance extracted was close to or exceeded 0.50, demonstrating good convergent validity and reliability among the five constructs. The Heterotrait-Monotrait (HTMT) ratios were below 0.85, indicating good discriminant validity between the constructs.”

- Minor note: the symbol on line 240 is a dash (-) and not a minus (-).

We have amended as suggested on page 12 (header of Table 5).

Thank you for the opportunity to provide input. I hope the authors will find it useful and constructive.

We thank the reviewer for taking the time to review our manuscript.

Round 2

Reviewer 1 Report

The authors have well addressed most of my previous comments. There is one comment that I would like to follow up: 

The authors indicated that only second-order model was examined in the CFA without examining other possible structures such as the bifactor model and first order. To ensure the scientific rigor of the analysis, I would suggest the author to present and compare the model fit of these different competing models to determine the optimal model. If not, strong justification in the manuscript is expected.

Author Response

We have conducted a bifactor analysis as suggested by the reviewer. The methods are delineated on page 4, paragraph 4 in the statistical analysis subsection, while the results are presented on page 12, paragraph 2.

“... Bifactor analysis was also conducted to determine the dimensionality of the study tool. whereby the percent of uncontaminated correlations (PUC) and explained common variance (ECV) were examined with cut-off values of >0.70 respectively. …”

“In the bifactor analysis, while the percent of uncontaminated correlations (PUC) at 0.77 exceeded the cut-off value of 0.70 for unidimensionality, the explained common variance (ECV) at 0.67 falls just below the cut-off value. While this indicates that the unidimensional model is a possibility, taken together with the results from the parallel analysis in the EFA, as well as the from the CFA with the PLS-SEM model, the second-order model is the most optimal model for the NeAT-BC.”

We have also added a sentence on the need to further validate the dimensionality of the study tool with a larger sample size on page 16, paragraph 3 in the study strengths and limitations section.

“… The dimensionality of the NeAT-BC should also be further explored in future studies with a larger patient sample. …”

Round 3

Reviewer 1 Report

Authors have thoroughly address my concerns.